# Software-Automatized Individual Lactation Model Fitting, Peak and Persistence and Bayesian Criteria Comparison for Milk Yield Genetic Studies in Murciano-Granadina Goats

**María Gabriela Pizarro Inostroza** [1,2]**, Francisco Javier Navas González** [1,]*****, Vincenzo Landi** [3]**, José Manuel León Jurado** [4]**, Juan Vicente Delgado Bermejo** [1]**, Javier Fernández Álvarez** [5] **and María del Amparo Martínez Martínez** [1]

[1] Department of Genetics, Faculty of Veterinary Sciences, University of Córdoba, 14071 Córdoba, Spain; kalufour@yahoo.es (M.G.P.I.); id1debej@uco.es (J.V.D.B.); amparomartinezuco@gmail.com (M.d.A.M.M.)

[2] Animal Breeding Consulting, S.L., Córdoba Science and Technology Park Rabanales 21, 14071 Córdoba, Spain

[3] Department of Veterinary Medicine, University of Bari "Aldo Moro", 70010 Valenzano, Italy; vincenzo.landi@uniba.it

[4] Centro Agropecuario Provincial de Córdoba, Diputación Provincial de Córdoba, 14071 Córdoba, Spain; jomalejur@yahoo.es

[5] National Association of Breeders of Murciano-Granadina Goat Breed, Fuente Vaqueros, 18340 Granada, Spain; j.fernandez@caprigran.com

***** Correspondence: fjng87@hotmail.com; Tel.: +34-651-679-262

**Abstract:** SPSS model syntax was defined and used to evaluate the individual performance of 49 linear and non-linear models to fit the lactation curve of 159 Murciano-Granadina does selected for genotyping analyses. Lactation curve shape, peak and persistence were evaluated for each model using 3107 milk yield controls with an average of 3.78 ± 2.05 lactations per goat. Best fit (Adjusted $R^2$) values (0.47) were reached by the five-parameter logarithmic model of Ali and Schaeffer. Three main possibilities were detected: non-fitting (did not converge), standard (Adjusted $R^2$ over 75%) and atypical curves (Adjusted $R^2$ below 75%). All the goats fitted for 38 models. The ability to fit different possible functional forms for each goat, which progressively increased with the number of parameters comprised in each model, translated into a higher sensitivity to explaining curve shape individual variability. However, for models for which all goats fitted, only moderate increases in explanatory and predictive potential (AIC, AICc or BIC) were found. The Ali and Schaeffer model reported the best fitting results to study the genetic variability behind goat milk yield and perhaps enhance the evaluation of curve parameters as trustable future selection criteria to face the future challenges offered by the goat dairy industry.

**Keywords:** goodness of fit; linear and nonlinear regression; mathematical modeling; parametric models; shape of lactation curve

## 1. Introduction

When research involves excessively high costs, researchers may be compulsorily forced to perform sample selection procedures [1]. These procedures seek to achieve the highest representativity of the population under study in the minimum possible number of effective

individuals. Limited samples are a common element of research whose objective is to determine the genetic background which regulates the expression of economically important traits.

The effects of sample size limitation become determinant when traits are obtained after the application of functions to model the trends that such traits describe (as happens in milk yield, composition or growth, among others). In such circumstances, sample size limitations may not only compromise population representativeness, but also may reduce the buffer effect derived from the dilution of the loss of information when larger numbers of individuals are considered. In this context, the application of general 'best fitting' models considered for milk yield standardization and composition may become impractical [2]. Hence, it may not be able to represent the reality of the populations under study.

Although numerous empirical linear and nonlinear parametric functions have been considered to modelize standardized lactation curves in animal populations at a large scale [3], some functions may be preferred over others. Concretely, these functions may differ in regards to their mathematical properties, computational complexity, the number of parameters that they comprise, the degree of relationship that they present with the main features of a typical lactation cycle or their ability to fit a wider range of curves. As a result, best fitting function (model) selection may depend on the higher or lesser ability of each certain function to report specific information on the milk yield outcomes of a certain individual or population.

The fitting properties of common models can be regularly consulted across the scientific literature from decades ago. However, reaching consensus on a standardized model able to accurately fit the maximum number possible of curves, that is, which is able to capture the most of the variability for most of the situations found in the field, is a challenging task to accomplish [4].

From a scientific perspective, the idea of a general model applied to individual goat lactations becomes even more striking, provided that the occurrence odd fitting curves in percentages that reach up to 6% in the Murciano-Granadina breed are described by from four (highly selected populations in stable environments) to 32 possible theoretical groups when combinations of the signs of the parameters of each model are determined in other breeds [2].

Contextually, individualized multimodel fitting may improve the outcomes obtained when fitting the standardized models to whole populations or samples. This, in turn, may depict the situation found in farms rather accurately. However, the process is often time-consuming and ineffective, and it may end up reporting inaccurate outcomes for curve shape parameters.

In an attempt to save these drawbacks, the literature has reported the use of software such as Wombat or ASREML with mixed models to automatize the process of issuing equations of lactation curves in genetic analyses [5]. Alternatively, statistic software such as SPSS or SAS is readily available and can be relatively easily used for the modellization of lactation curves without the need for extended specific knowledge [6].

Comparing the effectiveness of different models over individual studies may be a difficult task to develop, due to the idiosyncratic nature of milk supply modeling [7]. However, mixed models are, by definition, linear and can combine population and animal-specific effects. As a result, mixed models can be used to model specific data seeking to predict milk yield for a certain animal or herd in the unique conditions to which they may be exposed [7]. In this context, the comparative evaluation of certain nonlinear random and auto-regressive models [7,8] has been reported to be capable of increasing prediction accuracy and especially suitable for short-term milk-yield predictions [7].

Even if the use of SPSS software (Armonk, NY, USA) does not require extensive knowledge in the operators and may help save time, model syntax is not often found or accessible in the literature and requires certain computational skills. This framework makes the mechanization of the computer-based stages of the process difficult, which is not time- or resource-effective. This situation becomes even more evident when the latter aim is to implement individualized methods at large-scale population levels.

The description of the methods to perform individualized evaluation of the goodness of fit may prevent the occurrence of problems derived from drastically fitting biologically atypical lactation

curves to standardized models. As a result, many practical purposes may benefit from the study of individual patterns, such as health monitoring, individual feeding, and, especially in recent years, its application in breeding and genetic evaluations [9].

In these regards, standardized modellization of lactation curves may provide summary information, which may be determinant when making management and breeding decisions. That is, mathematical parameters determining milk yield prediction, and describing curve scale (initial and maximum yields), shape (time of maximum yield and persistence) or its biological/economic importance may become suitable candidates to be considered as potential new breeding criteria [10].

To this aim, once model syntax for the SPSS software for forty-nine models found in the literature to fit lactation curves has been described, the first objective of this paper was to determine the explanatory ability and predictive potential of each of the models defined. Second, the outcomes of lactation curve shape parameters (peak yield and persistency), model fitting properties and parameter estimation were compared in a sample of Murciano-Granadina goat selected to perform genotyping studies using Bayesian methods. As a result, the identification of the best fitting model for individual lactation curves may enable a more realistic comparison of individual curves and parameters. This model may not only better adapt to the mathematical nature described by each lactation curve itself, but may consider the particular situation of each animal at the moment such animals were evaluated.

## 2. Materials and Methods

### 2.1. Animal Sample and Sample Selection Process

The individuals registered in the studbook of the National Association of Breeders of Goats of Murciano-Granadina breed (CAPRIGRAN) were ranked, considering the official breeding value for milk yield and content that they obtained at the latest genetic evaluation at the time of sampling (published in stud catalog in 2015). A total of 159 herdbook-registered (Delgado et al., 2005) individuals were considered in the analysis. Animals in the sample belonged to 28 farms in the South of Spain, whose records were collected in random periods, from 2005 to 2018. The minimum age in the range was 1 year, the $Q_1$ age was 1.24 years, the median age was 1.35 years, the $Q_3$ age was 1.50 years, and the maximum age was 9.15 years.

### 2.2. Milk Performance Standardization

Murciano-Granadina is a polyestric breed. Its husbandry practices consider two kidding seasons each year, with lactation periods ranging from 210 to 240 days [11]. Total milk yield was estimated until 210 lactation days and expressed in Kgs as described in Pizarro, et al. [12] following the technique applied by CAPRIGRAN, provided the methods proved to be as accurate as the Fleischmann method, as suggested in the guidelines in ICAR [13].

Milk yield for each goat was computed through real production ($RP_j$) following the equation

$$RP_j = d_1 P_1 + 30 \sum_{i=n}^{n_j - 1} Pi_j + \left[ d_2 - 30(n_j - 2) \right] Pn_j$$

where $RP_j$ is real production of the $j$th goat; $P_1$ is milk yield at first control; $n$ is the number of controls; $Pi_j$ is milk yield in ith control $i$ for $j$th goat, $Pn_j$ is milk yield at the last control for $j$th goat.

Official control procedure is described in the Royal Decree Law 368/2005, of 8th April 2005 and milk performance recordings were performed at each farm according to the ICAR protocol (AT4, AT4T, AT4M, A6, AT6M, or AT6T) chosen by the farmer. The first control and the last, which were assessed individually for each goat computing the days ($d_1$) between kidding date (KD) and the date of the first control (FC), using the formula $d_1 = FC - KD$, and the days between the penultimate control (PC) and the last control (LC), using $d_2 = LC - PC$.

Lactation yields were then standardized/normalized to provide a reasonably equitable comparison of dairy goats with different lactation characteristics as suggested in Norman, et al. [14].

Normalized milk yield per each goat at 210 days was calculated using the formula $NP_j = d_1 P_1 + A + B$, where $NP_j$ is the normalized yield for goat j. A and B could be defined as; $A = 30 \sum_{i=1}^{n_j-2} \frac{P_i P_j + 1}{2}$ , $B = \left[d_2 - 30(n_j - 2)\right] \frac{Pn_j - 1 + Pn_j}{2}$.

The model used to calculate normalized yields at 210 days is described by $MP210 = \sum_{i=1}^{n-1} \left[ \left( \frac{pldc_i + pldc_{i+1}}{2} \right) \cdot I_{i\,i+1} \right]$, from which MP210 is the accumulated milk yield until 210 lactation days; $pldc_i$ is milk yield during milk control $i$; $pldc_{i+1}$ is milk yield in the following milk control and $I_{i,i+1}$ is the day interval between two consecutive controls.

## 2.3. Milk Production Records

A total of 3107 milk yield control records from 399 lactations (average of 3.91 ± 2.01 lactations per goat) belonging to 159 genotyped goats were considered in the statistical analyses. Days from parity to first control were, on average, 21.21 ± 13.71. Number of controls per lactation were, on average, 4.80 ± 2.86. Parametric assumptions (normality and homoscedasticity) were tested on our study sample to determine whether distribution properties could have been biased as a result of the process of sample selection. The Shapiro–Francia test routine of the Test and distribution graphics package of the Stata Version 15.0 software process was used to test the normality. Levene's test to test variance homogeny of variance across groups (homoscedasticity) of the SPSS Statistics for Windows statistical program, Version 25.0.

## 2.4. Lactation Curve Models and Curve Shape Parameters

Forty-nine linear and non-linear models were used to describe the lactation curves for milk yield of the 159 considered in this study. The equations for these models are presented in Table S1. Table S1 also presents the code that we will use to refer to each model across the manuscript and literature referencing works in which every model was applied. Linear and non-linear functions were used to model the relationship between milk yield and days in milk. As a way to facilitate the automatized application of the models in this study, Table 1 presents SPSS Model syntax. The syntax formulas defined in this paper are ready to be copied and pasted in the non-linear regression task from the Regression procedure of SPSS version 25.0 [15].

**Table 1.** SPSS models syntax for lactation curve in SPSS.

| Model Name | SPSS Model Syntax |
|---|---|
| Ali and Schaeffer model (ALISCH) | b0 + b1 * days + b2 * (days ** 2) + b3 * (lg10 (1/days)) + b4 * (lg10 (1/days) ** 2) |
| Asymptotic Regression, Single Exponential decay to an arbitrary value (SXPDCY) | b0 * (1 − b1) ** days |
| Asymptotic Regression, Lactation modification of Metcherlich Law of Diminishing Returns or Exponential growth model (METLAW) | b0 * (1 − b1 * exp (−b2 * days) − (b3 * days)) |
| Brody (BRODY) | b0 * Exp (−b1*days) − b0 * Exp(−b2*days) |
| Cappio Borlino, biexponential (CAPBOR) | b0 * days ** b1 * Exp(−b2 * days) |
| Cobby and Le Du (COBLDU) | b0 *1 − Exp(−b2 * days) − (b1 *days) |
| Compound/Exponential Growth (CEXPGR) | b0 * (b1 **days) |
| Cubic (CUBIC) | b0 + (b1 * days) + (b2 * days **2) + (b3 * days **3) |
| Cubic Spline function with one knot (CUBSPL) | b0 + b1 * days + b2 * days ** 2 + b3 * (days) ** 3 + b4 * (days − Knot) ** 3 |
| Curve S (CURVES) | Exp (b0 + (b1/days)) |

| | |
|---|---|
| Density (DENSITY) | $(b0 + b1 * days) ** (-1/b2)$ |
| Dhanoa (DHANOA) | $b0 * days ** (b1 * days) * Exp(-b2 * days)$ |
| Dijkstra (DJKSTR) | $b0 * Exp(b1 * (1 - Exp(-b2 * days)) b2) - b3 * days$ |
| Exponential decline function or Gaines (EDFGAIN) | $b0 * Exp(-b1 * days)$ |
| Gauss (GAUSS) | $b0 * (1 - b2 * Exp(-b1 * days ** 2))$ |
| Gompertz (GMPRTZ) | $b0 * Exp(-b1 * Exp(-b2 * days))$ |
| Grossman (GROSMN) | $b0 * (days ** b1) * Exp(-b2 * days) * (1 + b3 * SIN(days) + b4 * Cos(days))$ |
| Hayashi (HAYSHI) | $b1 * Exp(-b2/days - Exp(-days/b0 * b2))$ |
| Inverse quadratic polynomial (INVQPOL) | $days * (b0 + (b1 * days) + (b2 * (days ** 2))) ** (-1)$ |
| Inverse, Linear hyperbolic (INVLINHY) | $b0 + (b1/days)$ |
| Johnson Schumacher (JOHNSCH) | $b0 * Exp(-b1/(days + b2))$ |
| Log Logistic (LOGLOG) | $b0 - ln(1 + b1 * Exp(-b2 * days))$ |
| Log Modified Weibull (LGMWEIB) | $(b0 + (b2 * days)) ** (b1)$ |
| Logarithmic (LOGARITH) | $b0 + (b1 * ln(days))$ |
| Madalena (MADALN) | $b0 - b1 * days$ |
| Michaelis Menten (MICHMEN) | $b1/days * (1 + (b2/210) ** b1)/(1 + (b2/days) ** b1) * (1 + (days/b2) * b1)$ |
| MilkBot (MILKBOT) | $b0 * (1 - (Exp((b2 * days)/b1))/2) * Exp(-b3 * days)$ |
| Molina and Boschini/Modal Linear (MOL & BOS) | $b0 - b1 * Abs(days - (b2))$ |
| Morgan Mercer Florin (MORMFLO) | $((b0 * b1) + (b2 * days ** b3))/(b1 + (days ** b3))$ |
| Nelder, inverser polynomial, Yadav (NELDER) | $days/(b0 + b1 * days + b2 * days ** 2)$ |
| Parabolic exponential and Parabolic, Sikka (PEMSIK) | $b0 * Exp((b1 * days) - (b2 * days ** 2))$ |
| Parabolic yield-density (PARYLDENS) | $(b0 + (b1 * days) + (b2 * (days ** 2))) ** (-1)$ |
| Power (POWER) | $b0 * (days ** b1)$ |
| Quadratic cum log (QDCMLOG) | $b0 + b1 * days + b2 * days ** 2 + b3 * ln(days)$ |
| Quadratic (QUADRT) | $b0 + (b1 * days) + (b2 * days ** 2)$ |
| Quadratic model Dave (DAVE) | $b0 + b1 * days - b2 * days ** 2$ |
| Quadratic spline function with one knot (QUADSPL) | $b0 + b1 * days + b2 * days ** 2 + b3 * (days - Knot) ** 2$ |
| Ratio Cubics/Partial Fraction with Cubic Denominator (RATCUB) | $(b0 + b1 * days + b2 * days ** 2 + b3 * days ** 3)/(b4 * days ** 3)$ |
| Ratio Quadratics/Partial Fraction with Quadratic Denominator (RATQUAD) | $(b0 + b1 * days + b2 * days ** 2)/(b3 * days ** 2)$ |
| Richards (RICHRDS) | $b0/((1 + b2 * Exp(-b1 * days)) ** (1/b3))$ |
| Rook (ROOK) | $b0 * (1/1 + (b1/b2 + days)) * Exp(-b3 * days)$ |
| Simple Linear (SIMLIN) | $b0 + (b1 * days)$ |
| Singh And Gopal (SIN&GOP) | $b0 - b1 * days + b2 * ln(days)$ |
| Third order Legendre orthogonal polynomial (3ORDLEG) | $b0 * 0.7071 * (2 * ((days - 1)/(210 - 1)) - 1) ** 0 + (b1 * 1.2247 * (2 * ((days - 1)/(210 - 1)) - 1) ** 1) + ((b2 * -0.7906 * (2 * ((days - 1)/(210 - 1)) - 1) ** 0) + (2.3717 * (2 * ((days - 1)/(210 - 1)) - 1) ** 2)) + ((b3 * -2.8062 * (2 * ((days - 1)/(210 - 1)) - 1) ** 1) + (4.6771 * (2 * ((days - 1)/(210 - 1)) - 1) ** 3))$ |

| | |
|---|---|
| Verhulst/Logistic differential equation/Pearl Reed (VERHLST) | b0/(1 + b1 * (Exp (−b2 * days))) |
| Von Bertalanffy (VBRTLNFY) | b0 * (1 − (b1) *(Exp(−b2 * days))) ** 3 |
| Weibull, Parametric Survival Models (PARSURW) | b0 − (b1 * (Exp (−b2 *(days ** b3)))) |
| Wilmink's exponential (WILMINK) | b0 + b1 * Exp (−0.05 * days) + b2 * days |
| Wood (WOOD) | b0 − b1 * Exp(−0.05 * days) − b3 * days |

Days: days in milk.

The initial search grid was specified covering the parameter bounds of each model (b0, b1, b2, b3 and b4 parameters). An iterative process using the curve estimation task from the Regression procedure of SPSS version 25.0 [15] was used. The iterative process considered for as many rounds as was necessary until a tolerance convergence criterion of $10^{-8}$ was reached, as suggested by other authors, as stricter criteria such as $10^{-6}$ or $10^{-8}$ have been suggested to report the same outcomes out of a slightly higher number of iterations [16]. Convergence criterion was defined as the error sum of squares between successive iterations. Once determined, initial parameters were pre-set and considered to run the mechanized protocols for model fitting. The Levenberg–Marquardt method was used as the default iteration method. A maximum of 2000 iteration rounds were used for each analysis, as suggested in IBM SPSS Statistics Algorithms version 25.0 by IBM Corp. [17]. Average number of rounds to achieve convergence criterion was $3.158 \pm 0.682$ ($\mu \pm$ SD).

*2.5. Model Selection Criteria*

As suggested by Tedeschi [18], evaluating model suitability to predict or describe the trends in data from the field can only be achieved if several statistical analyses are combined and interpreted. The use of only a few techniques may be misleading in selecting the appropriate model in a given scenario. Residual values are computed after the result from the difference between observed value and predicted values. The Shapiro–Francia test was run on the residuals of each model to determine whether they are normally distributed or not. The Durbin–Watson statistic tests the null hypothesis that the residuals from an ordinary least-square regression are not auto-correlated against the alternative that the residuals follow an auto-regressive process. The Durbin–Watson statistic ranges from 0 to 4. The Durbin–Watson test is reliable for sample sizes larger than 15 [19]. Durbin–Watson statistics are only suitable for ordered time or spatial series [20]. A value nearer to two indicates non-auto-correlation, a value near 0 indicates positive auto-correlation and a value near 4 indicates negative auto-correlation. The Durbin–Watson test [21] was conducted on the residuals of each model (using mean daily yields of each day of lactation) to test for possible first-order autocorrelations among residuals. Positive autocorrelations occur when adjacent values of residuals tend to share the same sign (positive or negative) more than is randomly possible. The Linear regression test of the regression procedure in SPSS version 25.0. provided the Durbin–Watson statistic.

Among the accuracy and precision criteria suggested by Tedeschi [18], we chose those parameters, which were rather appropriate and common for lactation curve model comparison. Model selection criteria included percentage of successfully fitted lactation curves (Table 2), RSS, MSPE, Adjusted R Squared (Adj. $R^2$), Akaike information criterion (AIC), corrected Akaike information criterion (AICc) and Bayesian information criteria (BIC) (Table S3).

**Table 2.** Mean Adjusted coefficient of determination (Adj. $R^2$), Adj. $R^2$ Standard deviations (SD) and percentage of successfully fitted lactation curves of the models of lactation curve for milk yield (Kg) in goat Murciano-Granadina.

| Model | μ Adj. $R^2$ | SD Adj. R | Percentage of Successfully Fitted Lactation Curves |
|---|---|---|---|
| Ali and Schaeffer model (ALISCH) | 0.469 | 0.236 | 100.00% |
| Asymptotic Regression, Single Exponential decay to an arbitrary value (SXPDCY) | 0.143 | 0.182 | 73.53% |
| Asymptotic Regression, Lactation modification of Metcherlich Law of Diminishing Returns or Exponential growth model (METLAW) | 0.301 | 0.210 | 100.00% |
| Brody (BRODY) | 0.304 | 0.214 | 100.00% |
| Cappio Borlino, biexponential (CAPBOR) | 0.347 | 0.223 | 100.00% |
| Cobby and Le Du (COBLDU) | 0.276 | 0.215 | 100.00% |
| Compound/Exponential Growth (CEXPGR) | 0.080 | 0.003 | 9.80% |
| Cubic (CUBIC) | 0.440 | 0.232 | 100.00% |
| Cubic Spline function with one knot (CUBSPL) | 0.440 | 0.232 | 100.00% |
| Curve S (CURVES) | 0.137 | 0.168 | 100.00% |
| Density (DENSITY) | NC | NC | NC |
| Dhanoa (DHANOA) | 0.345 | 0.232 | 100.00% |
| Dijkstra (DJKSTR) | 0.374 | 0.201 | 100.00% |
| Exponential decline function or Gaines (EDFGAIN) | 0.194 | 0.182 | 100.00% |
| Gauss (GAUSS) | NC | NC | NC |
| Gompertz (GMPRTZ) | NC | NC | NC |
| Grossman (GROSMN) | 0.424 | 0.222 | 100.00% |
| Hayashi (HAYSHI) | 0.209 | 0.201 | 100.00% |
| Inverse quadratic polynomial (INVQPOL) | NC | NC | NC |
| Inverse, linear Hyperbolic.(INVLINHY) | 0.135 | 0.163 | 100.00% |
| Johnson Schumacher (JOHNSCH) | 0.237 | 0.211 | 100.00% |
| Log Logistic (LOGLOG) | NC | NC | NC |
| Log Modified Weibull (LGMWEIB) | NA | NA | 1.96% |
| Logarithmic (LOGARITH) | 0.186 | 0.194 | 100.00% |
| Madalena (MADALN) | 0.195 | 0.182 | 100.00% |
| Michaelis Menten (MICHMEN) | NC | NC | NC |
| MilkBot (MILKBOT) | NA | NA | 9.80% |
| Molina and Boschini/Modal Linear (MOL & BOS) | 0.255 | 0.194 | 100.00% |
| Morgan Mercer Florin (MORMFLO) | 0.275 | 0.242 | 100.00% |
| Nelder, inverser polynomial, Yadav (NELDER) | 0.441 | 0.233 | 100.00% |
| Parabolic exponential model and Parabolic, Sikka (PEMSIK) | 0.331 | 0.230 | 100.00% |
| Parabolic yield-density (PARYLDENS) | NC | NC | NC |
| Power (POWER) | 0.178 | 0.187 | 100.00% |
| Quadratic cum log model (QDCMLOG) | 0.420 | 0.231 | 100.00% |
| Quadratic (QUADRT) | 0.210 | 0.230 | 100.00% |
| Quadratic model Dave (DAVE) | 0.328 | 0.230 | 100.00% |
| Quadratic spline function with one knot (QUADSPL) | 0.328 | 0.230 | 100.00% |
| Ratio Cubics/Partial Fraction with Cubic Denominator (RATCUB) | 0.340 | 0.228 | 100.00% |
| Ratio Quadratics/Partial Fraction with Quadratic Denominator (RATQUAD) | 0.246 | 0.211 | 100.00% |

| | | | |
|---|---|---|---|
| Richards (RICHRDS) | NA | NA | 7.84% |
| Rook (ROOK) | 0.314 | 0.233 | 100.00% |
| Simple Linear (SIMLIN) | 0.195 | 0.183 | 100.00% |
| Singh And Gopal (SIN & GOP) | 0.338 | 0.223 | 100.00% |
| Third order Legendre orthogonal polynomial (3ORDLEG) | NC | NC | NC |
| Verhulst/Logistic differential equation/Pearl Reed (VERHLST) | 0.210 | 0.196 | 100.00% |
| Von Bertalanffy (VBRTLNFY) | 0.225 | 0.206 | 100.00% |
| Weibull, Parametric Survival Models (PARSURW) | NC | NC | NC |
| Wilmink's exponential (WILMINK) | 0.342 | 0.227 | 100.00% |
| Wood (WOOD) | 0.342 | 0.227 | 100.00% |

NC: The model does not converge; NA: Not all animals converged, hence it was not possible to compute it.

Residual sum of squares (RSS) is a statistical technique used to measure the amount of variance in a dataset that is not explained by a regression model. If we consider a regression to be a measurement of the strength of the relationship between a dependent variable and an independent variable in a set of independent variables, then the RSS measures the amount of error remaining between the regression function and the dataset. A smaller RSS figure represents a regression function. This essentially determines how well a regression model explains or represents the data in the model.

Additionally, although Mean Square Residual or Error (MSE) have been used and widely reported to measure how close a regression line is to a set of points, that is, how well a certain model fits the data being observed and Minimum Mean-Square Residual or error (MMSE), mean square prediction error or MSPE (=RSS/no. of observations) was chosen to measure error variation given that MSE has been reported to be influenced by the number of parameters [22] in cases of reduced sample sizes like those in genotyping studies.

R squared ($R^2$) is a biased measure of the proportion of the variance explained by the model (from 0 to 1 = 0 to 100%), as the more terms are added into the model as predictors, the more it increases, causing overfitting in models with many parameters as it amplifies the border effect of polynomials. Hence, adjusted R squared or modified R squared (Adj. $R^2$) was used given that it compensates for such overfitting by penalizing for the number of terms included as predictors.

The coefficient of determination lies always between 0 and 1, and the fit of a model is satisfactory if $R^2$ is close to unity. Contrastingly, negative Adjusted $R^2$ appears when the Residual sum of squares approaches to the total sum of squares, which means the explanation towards response is very low or negligible. Hence, negative Adj. $R^2$ may mean the insignificance of explanatory variables. However, results may be improved as sample size increases.

The ratio of Adj. $R^2$ to $R^2$ is measured from 0 to 1 and accounts for the likely decrease in model fit when a certain model is applied to new data. The higher the ratio of Adj. $R^2$ to $R^2$, the less affected by overfitting the model will be. In this context, ideally, Adj. $R^2$ should be as much close to $R^2$ as possible for a good fit, which would also mean potential overfitting may have been considered and quantified. A ratio from 0 and 0.4 indicates severe overfitting problems.

Following the premises of information theory, several methods have been described to compare models as regards their ability to explain or capture the variability observed in the dataset being studied (AIC and AICc) and the predictive potential (BIC) of the model designed for the data being modelled.

Akaike information criterion (AIC), Corrected Akaike information criterion (AICc) and Bayesian information criterion (BIC), were calculated as follows

$$AIC = Nln(R\frac{SS}{N}) + 2K$$

where $RSS$ is the residual sum of squares, $N$ is the number of datapoints and $K$ is the number of independent parameters of the model.

When a large number of observations (*N*) is not present, or for models containing a relatively large number of parameters, the corrected AICc may be more accurate, however, similar results of AIC and AICc are likely to be reported where a high number of observations is studied. AICc should be used when N/K < 40.

$$AICc = AIC + 2K \frac{(K + 1)}{N(N + 1)}$$

where *K* is the number of parameters and *N* is the number of observations.

Bayesian information criterion (BIC; [23]) is a model-order selection criterion and penalizes more complicated models for the inclusion of additional parameters and was computed after

$$BIC = N * N \, ln\left(\frac{RSS}{N}\right) + K * ln(N)$$

where *RSS* is the residual sum of squares, *N* is the number of observations or records and *K* is the number of independent parameters of the model.

*2.6. Bayesian Model Criteria Comparison*

As suggested by Tedeschi [18], the concordance correlation coefficient (CCC) measures if predicted values are precise and accurate. CCC is also known as a reproducibility index. According to the same authors [18], CCC is based on the Pearson's correlation coefficient estimate (*r*), which measures precision. Bayesian inference Pearson correlation function was used to characterize the posterior distribution of the linear correlation between predicted values of curve shape parameters (b0, b1, b2, b3 and b4) across models.

Additionally, Bayesian inference Pearson correlation function was used to characterize the posterior distribution of the linear correlation between curve shape parameters (b0, b1, b2, b3 and b4) in the same model. Correlation coefficients were analyzed to determine whether the values of some of these curve parameters could be related to other parameters in the curve, especially as model complexity increases. Bayesian inference for Pearson correlation was performed using the Pearson correlation task from the Bayesian statistics procedure in SPSS Statistics, Version 25.0, IBM Corp. (2017).

A full description of the algorithms used by SPSS to perform Bayesian Inference on Pearson correlation in this study can be found in the public document IBM SPSS Statistics Algorithms version 25.0 by IBM Corp. [17]. Once the relationship between curve shape parameters was determined, we evaluated whether model complexity could condition the better fitting properties of some models over others.

Additionally, considering that the size of the sample used in this study was small and sample distribution violated parametric assumptions, Bayesian inference for ANOVA was run to test for statistical differences in the mean for determination coefficient (scored through Adjusted $R^2$) and flexibility selection criteria (AIC, AICc and BIC) across models consisting of two, three, four or five regressors. This analysis was aimed at determining whether model complexity was a conditioning factor for the best fitting properties of variability capturing ability (Adj. $R^2$), observed data explanation (AIC, AICc) and predictive potential (BIC). Smaller numerical values of flexibility selection criteria (AIC, AICc, BIC) have been reported to be indicative of better fit properties when comparing models.

As suggested in public document IBM SPSS Statistics Algorithms version 25.0 by IBM Corp. [17], Bayesian inference of ANOVA is approached as a special case of the general multiple linear regression model. A full description of the algorithms used by SPSS to perform Bayesian Inference on Analysis of Variance (ANOVA) in this study can be found in the public document IBM SPSS Statistics Algorithms version 25.0 by IBM Corp. [17]. The tolerance value for the numerical methods and the number of method iterations were set as a default by SPSS v25.0 [15].

First, we interpreted the estimated effect of the factors considered in the predictive models, its interval and the posterior distribution statistics. The 95% Credibility Interval shows that there is a 95% probability that posterior distribution mean value for each factor in the population lies within

the corresponding intervals. When 0 is not contained in the Credibility Interval, a significant effect for such a factor is detected.

The Bayes factor (BF) measures the likelihood of null and alternative hypotheses or one model versus another based on the prior distribution and the data. It quantifies the change in the likelihood given in the prior to the posterior likelihood that is produced by the data. The BF is a measure of the strength of the evidence and is used instead of p values (from frequentist approaches) to reach a conclusion. A large BF implies that the evidence favours the alternative hypothesis compared to the null hypothesis, or of one model over the other.

Among all the priors suggested by the manual, the Jeffrey–Zellner–Siow mixture of g-priors was used for both Bayesian inference on Pearson's correlations and ANOVA. Jeffrey–Zellner–Siow's prior somehow appears as a data-dependent prior through its dependence on $X_i$, but this has been reported not to be a drawback since regression models are conditional on $X_i$. As suggested by Heck [24], JZS prior could be an alternative that may satisfy several theoretical requirements, such as the equality constraint on the test-relevant parameters, for instance β, which leads to the null hypothesis H₀ = β = β₀ [25]. The benefits of JSZ prior distribution were also reported by and Liang, et al. [26]. Contextually, conditional on the residual variance ($\sigma_{\varepsilon i}^2$), the JZS prior defines a multivariate Cauchy distribution for the slope parameters of the full model, as follows

$$(\beta_i | \sigma_{\varepsilon i}^2) \sim MVC(0_P, {\gamma_i}^2 \sigma_{\varepsilon i}^2 C_i^{-1}),$$

which is defined by a P-dimensional zero vector (location vector) and a scale matrix. The constant $\gamma_i$ determines the amount of scaling, which is chosen by the user a priori, the residual variance $\sigma_{\varepsilon i}^2$, and the matrix $C_i = X_i' X_i / N_i$, which is the covariance matrix of the centred design matrix $X_i$.

There are several qualities of the JZS prior which make it especially appropriate when performing ANOVA. Among others, the prior is symmetric and centered at zero, in line with the predictive matching criterion, as reported by Bayarri, et al. [27], hence positive and negative values of the slope parameters have a priori the same probability of occuring. Furthermore, JZS prior is scale-invariant, thus the resulting Bayes factor does not depend on the scale of both the dependent variable and factors or covariates, hence the results do not change when different unit variables are evaluated together, which is common in field conditions studies.

This independence from model regressors measurements is achieved by scaling the multivariate Cauchy distribution by the residual variance $\sigma_{\varepsilon i}^2$ (a priori, a larger residual variance implies larger slopes) and by the inverse of the covariance matrix $C_i$ (a priori, a covariate with a larger variance implies smaller slopes). It may be worth considering that the procedure of defining a scaled prior for unstandardized coefficients (βᵢ) equals to the process of defining a prior for standardized coefficients ($\beta_i^*$).

Third, the scale parameter $\gamma$ is fixed to a constant by the user, which allows prior beliefs to be specified about the expected effect size. The IBM Corp. [17] algorithm guide reports that the algorithm of JZS prior for linear regression analyses, to compute the Bayes Factor uses the default value of $\gamma = 2\sqrt{\pi} = 3.5$, which reflects a prior belief of a medium effect size. For a single covariate x, this choice implies that the standardized regression slope $\beta_i^* = \beta_i \cdot SD(x_i)/\sigma_i$ has an a priori probability of 53.2% of being in the range [−0.50, +0.50].

## 2.7. Peak and Persistency Computation for Best Fitting Model

Peak yield and persistency were computed following the premises proposed by the papers referenced in Table S1 and were specific to each of the functions considered. When provided with the nature of the best fitting model as suggested by statistical analyses, and when the computation of peak yield was not possible, change in variable units per event was computed as suggested in Table S2 and in Garson [28]. It has to be noted that theoretical definitions of persistency can be very different, but we decided to stay as close as possible to the definitions associated to the tested functions. Therefore, as Table S2 suggests, persistency could be computed differently depending on the model used as follows: descending rate of the curve after the lactation peak, rth relative rate of decline at the point halfway between peak yield and end of lactation or instantaneous rate of change.

When no specific function for these parameters was found in the literature, Symbolab® Mathematical calculation tool for education was used to determine relative maxima (peak yield) and descending rates in the curve depending on the model fitted (persistency).

## 3. Results

Table S4 shows a summary of the descriptive statistics for milk yield (Kg) records considered in this study, while Figure 1 shows a graphical representation of the evolution of milk yield through lactation across lactations (form first lactation to ninth lactation). The variation coefficient for milk yield (Kg) reported a value of 42.31%. Table 2 shows a summary of adjusted coefficients of determination (Adj. $R^2$) of the models for milk yield (Kg) lactation curve fitting in Murciano-Granadina goat. Adj, $R^2$ for the model reporting the best ability to capture variability was 0.469 for the model of Ali and Schaeffer model (ALISCH), while the minimum for Adj. $R^2$ values (0.80%) were reported for Compound/Exponential Growth (CEXPGR). Additionally, all goats converged for ALISCH, while the minimum fraction of goats converging for a specific model was 9.80% for CEXPGR (Table 2).

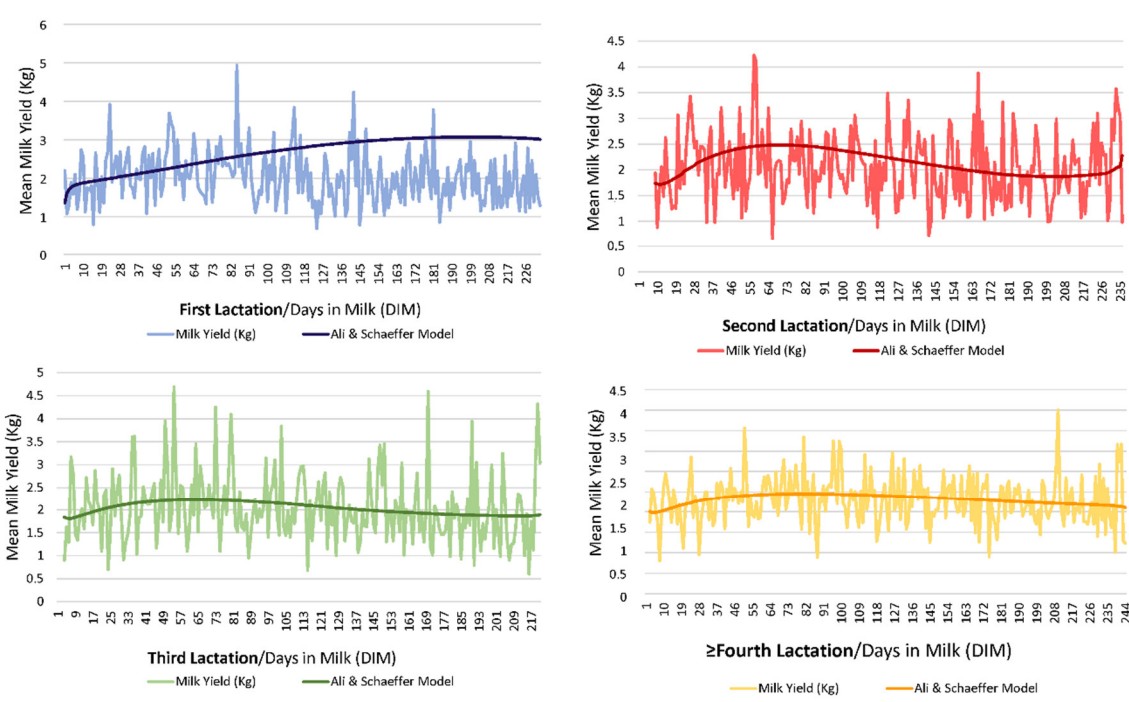

**Figure 1.** Graphical representation of the evolution of milk yield (Kg) through lactation from first, second, third and fourth or later lactations in Murciano-Granadina goats.

Parametric assumptions (normality, Shapiro–Francia test $p < 0.05$) and homoscedasticity, Levene's test, $p < 0.05$ across groups) were violated in our study dataset, hence we opted for the use of a nonparametric statistical alternative. As the sample used in this study was small, Bayesian analyses were run in an attempt to preserve model accuracy and power of the techniques applied.

Additionally, the Shapiro–Francia test was performed to test for residuals' normality, reporting statistically significant results for all fitted models ($p < 0.001$). Thus, residuals were not normally distributed. The Durbin–Watson statistic showed that all values were within the range of 0 to 2; thus, the residuals of all models were positively autocorrelated. The run test in our study indicated that the residuals of all models were not independent. These results are consistent with the earlier studies reported by Mohanty, et al. [29]

Table S3 shows a summary of the model curve shape parameters (b0, b1, b2, b3 and b4), number of model regressors, measures for model fit and flexibility selection criteria computed through residual sum of squares (RSS), mean square prediction error (MSPE), variability explanation power through Akaike Information Criterion (AIC) and Corrected Akaike Information Criterion (AICc) and Predictive power through Bayesian Information Criterion of the models that were used to fit Murciano-Granadina lactation curves. Almost all models reported a value for b0 around 2 as shown in Table S3, except for those implying a higher computational complexity, which, in fact, may have conditioned their better explicative and predictive potential (CURVS, CUBSPL, DJKSTR, INVQPOL, HAYSHI, LOGLOG, MILKBOT, NELDER, PARYLDENS, RATCUB, RATQUAD, ROOK, VERHLST, VBRTLNFY and 3ORDLEG).

Concretely, DENSITY, GAUSS, GMPRTZ, IVNQPOL, LOGLOG, MICHMEN, PARYLDENS, PARSURW and 3ORDEG failed to converge, hence no Adj. $R^2$ is reported for them. When b0 shape parameter came close to 0, flexibility selection criteria slightly increased. However, when the values for b0 highly differed from 0 in absolute value, a higher poor ability to explain and predict was suggested as shown in Table S3. With only a few exceptions, values for b1, b2, b3 and b4, were maintained around 0, except for the models that were reported above to have highly increased or highly decreased values of b0. Table S4 shows a summary of Bayesian inference Posterior Mean Distribution for Pearson's correlations between curve shape parameters (b0, b1, b2, b3, b4 and knot) across models.

The correlations between curve shape parameters (b0, b1, b2, b3 and b4), are presented in Table 3. A moderate evidence for the correlation between b0 and b1 and b3 was suggested by Pearson correlation Bayesian inference analysis (Table 3). However, the values for these correlations were negative and low. Higher positive and negative correlations were anecdotally evidenced between b2 and b4, respectively.

**Table 3.** Bayes factor inference on pairwise correlations among lactation curve parameters.

| | Parameters | $b_0$ | $b_1$ | $b_2$ | $b_3$ | $b_4$ |
|---|---|---|---|---|---|---|
| $b_0$ | Pearson Correlation | 1 | −0.017 | 0.256 | 0.031 | −0.317 |
| | Bayes Factor | | 8.709 | 2.453 | 5.263 | 2.644 |
| $b_1$ | Pearson Correlation | −0.017 | 1 | −0.047 | 0.039 | −0.401 |
| | Bayes Factor | 8.709 | | 7.533 | 5.243 | 2.451 |
| $b_2$ | Pearson Correlation | 0.256 | −0.047 | 1 | 0.040 | −0.580 |
| | Bayes Factor | 2.453 | 7.533 | | 5.093 | 1.907 |
| $b_3$ | Pearson Correlation | 0.031 | 0.039 | 0.04 | 1 | 0.999 |
| | Bayes Factor | 5.263 | 5.243 | 5.093 | | 0.037 |
| $b_4$ | Pearson Correlation | −0.317 | −0.401 | −0.58 | 0.999 | 1 |
| | Bayes Factor | 2.644 | 2.451 | 1.907 | 0.037 | |

Additionally, moderate evidence of correlation was found for b1 and b2 and b3 (though the value for these correlations were close to zero, almost equal in regards their significant value, but contrastingly, differed in sign, with b1 and b2 correlation being negative and b1 and b3 correlation being positive. The correlation between b1 and b4 was highly negative, but in this case, correlation found was only anecdotal. Correlation between b2 and b3 was moderately evidenced but poor, while correlation between b2 and b4 was anecdotally evidenced but highly negative. Contrastingly, there was anecdotal evidence of the lack of existence of a very high, almost complete correlation between b3 and b4.

Table 4 shows a summary of Bayesian ANOVA to test for differences in the mean for Adjusted $R^2$, AIC, AICc and BIC across models comprising two, three, four or five regressors. Significant differences were found for the mean of adjusted determination coefficient and flexibility selection criteria (AIC, AICc and BIC), when models comprised two, three, four or five regressors. An increasing trend was described with each element added to the model (0.10 to 0.12 points higher each time a new element was included). As regards flexibility selection criteria, the explicative and

predictive potential of models increased as the number of regressors considered in models decreased. However, the poorest values were reported for models involving three regressors (two plus the intercept). A poorer explicative and predictive performance was reported by models comprising an even number of regressors as opposed to those comprising an odd number of regressors.

**Table 4.** Summary of Bayesian ANOVA to test for differences in the mean for Adjusted $R^2$, AIC, AICc and BIC across models comprising two, three, four or five parametric regressors.

| | Adjusted $R^2$ | AIC | AICc | BIC |
|---|---|---|---|---|
| Sum of Squares | 0.206 | 181.408 | 181.309 | 181.376 |
| df | 3 | 3 | 3 | 3 |
| Mean Square | 0.069 | 60.469 | 60.436 | 60.459 |
| F | 5.792 | 0.427 | 0.426 | 0.427 |
| Sig. | 0.002 | 0.735 | 0.735 | 0.735 |
| Bayes Factor | 6.141 | 0.008 | 0.008 | 0.008 |
| 2 regressors models Posterior Mean | 0.152 | 64.936 | 69.510 | 62.173 |
| 2 regressors model 95% CI | 0.079–0.226 | 56.927–72.944 | 61.501–77.519 | 56.872–72.888 |
| 3 regressors models Posterior Mean | 0.266 | 68.766 | 73.337 | 68.713 |
| 3 regressors model 95% CI | 0.214–0.318 | 63.643–73.888 | 68.214–78.459 | 63.591–73.835 |
| 4 regressors models Posterior Mean | 0.273 | 64.635 | 69.205 | 64.583 |
| 4 regressors model 95% CI | 0.203–0.342 | 57.39–71.879 | 61.961–76.45 | 57.339–71.826 |
| 5 regressors models Posterior Mean | 0.418 | 64.928 | 69.498 | 64.878 |
| 5 regressors model 95% CI | 0.308–0.529 | 52.914–76.941 | 57.484–81.511 | 52.865–76.89 |

ALISCH was the best model not only with respect to its ability to capture population variability but also with regards to its explicative and predictive potential; this contrasts with our results for the comparison across the 49 models, as ALISCH involved five regressors, hence a higher parametric complexity could be presumed (Figure 2). This better performance could be attributed to the inclusion of logarithms in the model, as it was also reported for other models tested such as SIN and GOP, which, despite reporting a lower adjusted determination coefficient, presented a close value of flexibility selection criteria to those of the best fitting and performing models. In this case, model parametric complexity involved three regressors, which Table 4 had suggested to be the worst performing models on average with respect to explicative and predictive potential. Table 5 shows a summary of posterior distribution statistics and 95% credibility interval for milk yield (kg), peak yield and persistency.

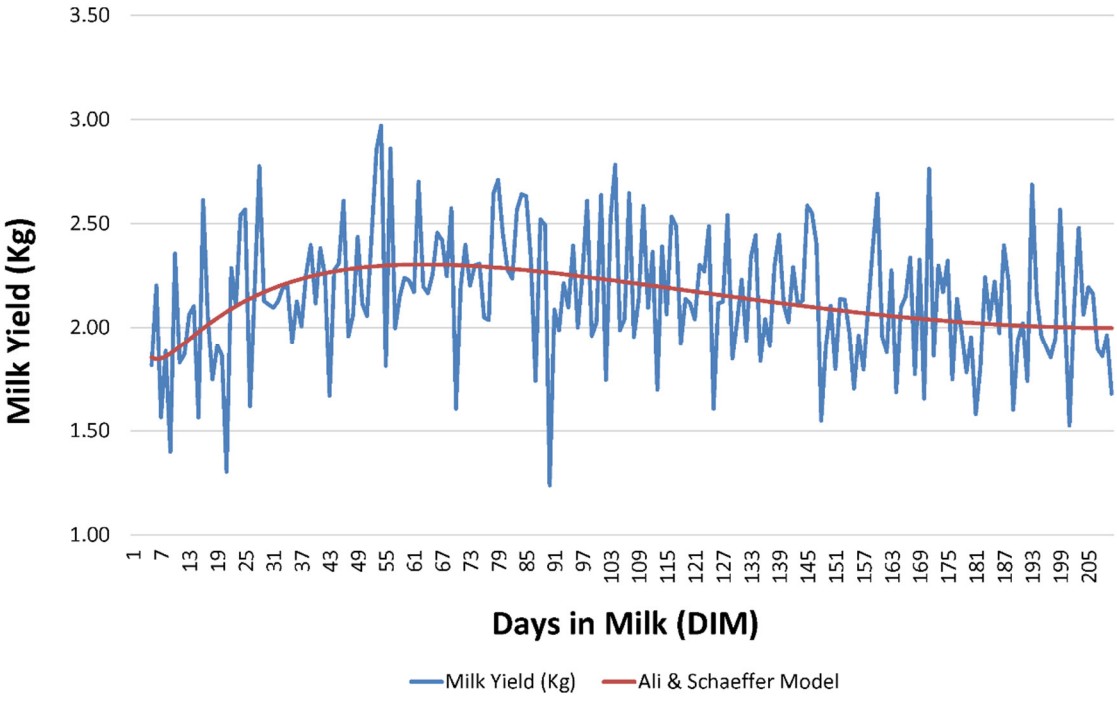

**Figure 2.** Graphical representation of the evolution of milk yield (Kg) and Ali and Schaeffer (ALISCH) model fit in Murciano-Granadina.

**Table 5.** Bayesian Posterior distribution estimates for milk yield (Kg), peak and persistence calculated for Ali and Schaeffer model (ALISCH) model in Murciano-Granadina goats.

| Parameter | Posterior | | | 95% Credible Interval | |
|---|---|---|---|---|---|
| | Mode | Mean | Variance | Lower Bound | Upper Bound |
| Milk yield (Kg) | 12.456 | 12.473 | 0.103 | 11.859 | 13.118 |
| Peak | 1311.836 | 1715.477 | 535,065.904 | 811.295 | 3561.277 |
| Persistence ($b_1$) | 0.031 | 0.040 | 0.000 | 0.019 | 0.083 |
| Persistence ($b_2$) | 0.000 | 0.000 | 0.000 | 0.000 | 0.000 |

## 4. Discussion

The relationship between variation and determination coefficient has been vaguely addressed in the literature. In these regards, in the linear dependence sense and in the context of our study, when the coefficient of determination is about 0.01, this means that only 1% of the variance in a measurement for milk yield could be explained by variation in time. The coefficient of variation (CV) shows the extent of variability in relation to the mean of the population [30]. Hence, if the ALISCH model reported an Adj. $R^2$ value of 46.90% in the context of a value of 42.31% for variation coefficient for milk yield, we could infer that ALISCH model may capture all the variability that may be attributed to the evolution of milk yield in time, plus around 4% of linear dependence with other factors. This may support the better explicative and predictive potential reported for this model when compared to the 48 models remaining. Still, our results provide slightly lower values for Adj. $R^2$ than other authors [2], which may be ascribed to the properties and characteristics of the sample that was used.

As suggested by Table 2, lower values of Adj. $R^2$ are related to lower percentages of successfully converging goats. Although ALISCH model could be considered a parametric complex model, due to the number of regressors that it comprises, our results suggest that the inclusion of logarithmic forms in the formula may somehow promote the adaptation of lactation curves described by each goat individually to the properties of the model, which may result in the improvement in the variability capturing ability of these models when compared to the rest.

González-Peña, et al. [2] suggested Ali and Schaeffer (ALISCH) [31] third-order Legendre's orthogonal polynomial (3ORDLEG) can detect or identify nine to fourteen curve shapes. Higher values for the correlations between estimated curve shape parameters values were reported when the Ali-Schaeffer model was compared to Legendre's polynomials. Thus, the differences across curve shapes that were identified when five-parameter models were applied could indeed be regarded as modifications of normal or abnormal curves, which are the two most frequent main forms of curve shape reported in the literature.

Brotherstone, et al. [32] comparatively evaluated the performance to model for random effects of orthogonal Legendre polynomials, the ALISCH function and two variations of the WILMINK function. These authors reached the conclusion that, independent of the conditioning phenotypical effect to be expected after the modelling features of each model and their fitting performance, the function used influences the estimates of genetic parameter for milk production. The same authors addressed the parametric functions of ALISCH and WILMINK, which were the ones reporting the best fitting capacity, even in the context of the negative correlations that were found between test-days at the beginning and the end of lactation. High values (Adj. $R^2$ 34.2%) were reported for Wilmink's exponential (WILMINK) function in the context of our results, with all goats successfully converging for the model. This may suggest that not only do the logarithmic forms included in computational methods promote model fit, but exponential forms also do.

Contrastingly, other authors such as [33] suggested that the fitting properties of orthogonal Legendre polynomials overcame those of ALISCH [31] and WILMINK functions to model random effects.

According to [34], the theoretical basis which supports these disagreeing results relies on the fact that model fit may be counteracted by a higher predictive and/or explicative error, given that, when using third-, fourth- or higher order polynomials to model, random regression models report erratic and implausible estimates of variance components, as a result of genetic parameters. This event is especially promoted when limited and/or unbalanced data are considered in modelling studies. For instance, when animals belonging to different age ranges unequally contribute to the dataset and/or when few records per animal are present in analyses in which the animals have fewer records than the order of the polynomials considered to model the records.

In line with these results, authors such as Pool and Meuwissen [35] suggested differences in regards to the goodness of fit depending on the order of the polynomial considered. This may be tightly related to the number of parameters estimated per animal but may also be limited by the computing capacity of the model used. Contextually, incomplete lactation records and heterogeneous milk yield variances may, therefore, require models with a higher parametric complexity (fifth-order of fit, etc.).

Spline functions can act as alternatives that can help reduce to the polynomials' degree. For this reason, spline functions have often been called segmented polynomials. Spline functions combine single segments of polynomials of low degree, which merge together in specific points which are called knots. Among the particularities of spline functions, they can be modeled using different bases which helps to minimize multicollinearity (depending on the method used), can be estimated easily given their linearity concerning the parameters that they involve, and are also easily and simply biologically interpretable.

Although our results for spline functions (QUADSPL and CUBSPL with one knot) reported among the highest values (44.0%) for the ability to capture variability (Adj. $R^2$), they did not outperform ALISCH model [31] in these regards. Thus, as flexibility selection criteria (AIC, AICc and BIC) were negligibly lower for CUBSPL with one knot, than for ALISCH or QUADSPL, respectively, the ALISCH model was still preferable when individualized lactation curves were to be fitted, due to their better ability to capture variability.

Ducrocq, et al. [36] suggested that the LGMWEIB model reported an overall adequate goodness of fit for the length of productive life data. However, some models reported an inadequate performance for records for which the 305 Mature Equivalent Milk Production (305ME) was not known. The same authors suggested this drawback should be easy to correct, as 305ME records can

be approximately predicted from early lactation tests. Additionally, Ducrocq, et al. [36] would also state that when such a correction is not feasible, for instance in cases of extremely short lactations, the corresponding length of productive life records should be assumed to be censored at the end of the previous lactation. These findings may support our results provided that the model did converge, but it did not reach good fitness values, which could be attributed to our reduced sample size and the specific characteristics of the lactations comprised in our study.

Gayawan and Ipinyomi [37] suggested, that in cases in which different models are chosen, a comparison of the ability of the competing models to reproduce the empirical data reveals that models chosen by $R^2$, are able to reproduce the data more appropriately than others, however, frequently, these models tend to be the most parametrically complex ones. This property would be completely extensible to Adj. $R^2$, in cases such as those, in which the model only uses one independent factor or covariate (days in milk).

Macciotta, et al. [9] suggested that problems related to the existence of different lactation curve shapes are usually neglected or solved drastically by considering shapes markedly different from the standard as biologically atypical. These authors reported that the meaning of parameters and the range of their values and of their correlations are clearly different among groups of curves. Our results suggest that curve parameters and their biological interpretation (peak yield and persistency) may as well vary across models, as computational methods have been addressed as different in the literature and as suggested in Table S2.

Additionally, our results contrast with those found in the literature for other species. For instance, the biological idiosyncrasies (time to peak, peak yield, number of peaks across lactation, total yield, value of persistence, among others) of the lactation of the different dairy species may condition the best fitting properties of certain models over the rest. For instance, cattle for which good outputs have been reported when the MILKBOT model was fitted [38], sheep and donkeys for which WOOD model has been suggested to fit well [39] and camels [40] for which the best model fit was found for fourth-order polynomials. Still, papers usually test the fitness ability of reduced numbers of models that have commonly been used in the literature. Hence, no direct comparison can possibly be made, as even if such models perform well, there is a lack of evidence of other potential models reporting better or worse results.

The literature suggests that the analysis of relationships between mathematical properties of models and lactation patterns should not only focus on the evaluation of fitting performances [9], as curve modeling usually deals with data of homogeneous groups of animals, and almost all proposed functions were able to fit average patterns at a more than acceptable level of accuracy. Contrastingly, two main issues may arise when a wide polymorphism can be found in regards to the specific curve shape parameters for each animal.

The first concern is the biological basis for the occurrence of different shapes or whether they may be conditioned by random perturbations like missing records or outliers. The second concern is related to the eventual differences that may occur both in the range of parameter values and in their mathematical interpretation when the same function is fitted to curves with different shapes.

Contextually, Macciotta, et al. [9], in their study of the second derivative of the Wood (WOOD) function, reported that the absolute value of the b parameter may control the magnitude of the deviation of the lactation pattern from a straight line. When b is positive, the curve is concave while when b is negative, it is convex. In standard curves, larger values of b may be related to a more rapid rate of increase of estimated yields in the first part of lactation, whereas in atypical curves increasing values of b may result in a reduced rate of decline in this phase.

Macciotta, et al. [9] also suggested that correlations may show higher absolute values for standard curves, resulting in a larger impact of b (b1 in our study) variations in the c (b2 in our study) parameter. These figures underline differences in the meaning of b (b1 in our study), hence its relevance in the determination of the atypical or typical nature of the curves fitted or the specific computation of the biological relationship between b and c and peak yield or persistency.

In models like WILMINK, a scaling factor which acts as a constant is included in both groups. Afterwards, differing between standard or typical curves and atypical curves, is the behavior of this

exponential term b(kt). The exponential term b(kt) increases in typical curves and decreases in atypical ones. As a result, the curve shape parameter b (represented by b1 in our study), with its asymptotic value for t being equal to 0, controls the rate of variation in the variable that we would like to measure (either it is milk yield or any of the components) in the first half of the curve, even when higher absolute values of b1 may indirectly result in faster increasing (or decreasing) rates.

Finally, the c parameter (b2 in our study, which quantifies the slope of the straight line ct), is directly related to the rate of decline in the second half of the curve (persistency of lactation or composition).

Low correlations between b1 and b2 (b and c), which describe both halves of the curve, revealed a substantial independence between the first and second part of the curve regardless of the type of curve considered (typical/standard and atypical curves). Contextually, the degree of independence between both halves of the curve across models is found in those differences, in which the basis for a greater or lesser ability to detect the standard curve could be supported.

Such differences in the ability of models to describe different lactation curve shapes was reported by Landete-Castillejos, et al. [41]. In line with their results, correlation values reported by our analyses may be supported by those in the study by Macciotta, et al. [9] as moderate evidence for the correlation between b0 (a) and b1 (b) and b3 (d) was suggested. However, the values for these correlations were negative and low. Higher positive and negative correlations were anecdotally evidenced between b2 (c) and b4 (e), respectively. Additionally, moderate evidence of low negative and positive correlations were found for b1 (b) and b2 (c) and b3 (d), respectively, which may also be supported by the aforementioned literature.

This framework suggests that two different kinds of complexity should be considered. On the one hand, parametric complexity or the number of effective parameters as an increasing trend (0.10 to 0.12 points higher each time) was observed to occur with each new element added to the model. On the other hand, the computational complexity (inclusion of operator different to the mere inclusion of the variables themselves) may mean increases in Adj. $R^2$ ranging from 0.10 to 0.30, when base 10 logarithms or exponential regressors are comprised in the models being fitted.

As regards flexibility selection criteria, the explicative and predictive potential of models decreased (AIC, AICc and BIC increased), as the number of regressors considered in models decreased (worst in parametric simpler models). However, the poorest values were reported for models involving three regressors (two plus the intercept) when compared to those models presenting less than three or over three parameters. This suggests that the model fitting decrease as a result of simpler models being used may be counteracted by the inclusion of complex computational regressors in the formulas. Simultaneously, a poorer explicative and predictive performance was reported by models comprising an even number of regressors as opposed to those comprising an odd number of regressors, but these may be attributed to random effects, as no reference has been found in these regards.

Relative predictive potential (assessed using Bayesian Information Criterion (BIC)) has been reported to be heavily dependent on the degree of unobserved heterogeneity between datasets, hence if heterogeneity is large, BIC will often perform better, due to the stronger penalty afforded [42]. Contrastingly, when heterogeneity is small, AIC or AICc will likely perform better.

Alternatively, our results that suggest values of AIC, AICc and BIC, due to their direct relationship with RSS and the number of observations/animals and parameters (through their cross relation as participating factors in their definition formulas), are normally attained to a similar proportional variation, and hence should not be determinant of a distinct model performance in regards to explicative or predictive potential. This could be attributed to individualized curve model fitting, as the specific treatment of the data belonging to each specific animal may mean the explication of intraindividual variability is maximized as much as possible in the context of the observations available for that particular animal.

As suggested by Brewer, et al. [42], the objective of the penalties implied by the flexibility selection criteria (AIC, AICc and BIC) is to reduce the effects of overfitting derived from the inclusion of a larger number of parameters. The same authors also reported that the penalty may be stronger

for BIC than AIC for any reasonable sample size. However, for small *n*, a corrected version AICc may provide a stronger penalty than AIC for smaller sample sizes, and stronger than BIC for very small sample sizes.

Since AICc is reported to have better small-sample behaviour, Burnham and Anderson [43] recommended the use of AICc as standard. The effect of a stronger penalty on the likelihood is to select smaller models, and so BIC tends to choose smaller models than AIC, and AICc also chooses relatively small models for smaller sample sizes. In consequence, BIC may tend, in realistic situations, to select models that are too simple (that is underfitted). This agrees with our results, although the relationship between complexity and flexibility selection criteria may not be strictly linear and may instead depend on the concepts of both computational and parametric complexity, with rather complex formulas acting as buffers for the decrease in the number of curve shape parameters.

In this context, ALISCH was the best model not only in respect to its ability to capture population variability, but also in regards to its explicative and predictive potential. This should be highlighted in the context of our results, as ALISCH involved five regressors, hence a higher parametric complexity could be presumed. As has been discussed above, the better performance of ALISCH could be attributed to the inclusion of logarithms in the model (higher computational complexity) rather than to the inclusion of a larger number of parameters, as was also reported for other models such as SIN and GOP. For instance, despite SIN and GOP reported a slightly lower adjusted determination coefficient, it presented a close value of flexibility selection criteria to those by best fitting and performing models (ALISCH among others). In this case, SIN and GOP parametric complexity involved three regressors, which Table 5 had suggested to be the worst-performing models in regards explicative and predictive potential.

Parametric complex models may benefit from the inclusion of logarithmic forms in their functions as this practice may promote the adaptation of lactation curves described by each goat individually to the properties of the model. In these contexts, spline functions (QUADSPL and CUBSPL with one known) do not outperform the ALISCH model. In the first 30–45 days of lactation, milk is often suckled by kids, then the first useful functional control can occur after the lactation peak or very close to it. Furthermore, goats are generally bred in extensive or semi-intensive systems. Under these conditions, the occurrence of a double peak, due to the high availability of pasture in the spring, has been widely reported [44]. Therefore, the relationship of early segments of goat lactations to total lactation milk yield has a high predictive value: for the first 69 days 68%, for 100 days of 87%, and for 140 days into lactation of 96%.

## 5. Conclusions

Parametric complex models may benefit from the inclusion of logarithmic forms in their functions as this practice may promote the adaptation of lactation curves described by each goat individually to the properties of the model. In these contexts, spline functions (QUADSPL and CUBSPL with one know,) do not outperform ALISCH model. Adj. R2 may be the best model selection parameter to consider especially in cases in which the model only uses one independent factor or covariate (days in milk). The differences between specific computational methods should be used to compute peak yield and persistency, which directly depend on curve shape parameters. Substantial independence between the first and the second part of the curve could be the reason for the larger number of standard curves detected across models. The relationship between complexity and flexibility selection criteria may not be strictly linear and may rather depend on the concepts of both computational and parametric complexity, with rather complex formulas acting as buffers for the increase in the number of curve shape parameters. The ALISCH model be preferable in studies in which reduced samples are used, and individualized automatized study of lactation curves for goat milk is performed, as it may capture all the variability that may be attributed to the evolution of milk yield in time, plus around 4% of linear dependence with other factors.

**Supplementary Materials:** The following are available online at www.mdpi.com/2227-7390/8/9/1505/s1, Table S1: Lactation curve model equations and where to find them in literature; Table S2: Peak yield and persistence estimates for each lactation curve model for milk yield and where to find them in literature; Table S3: Summary

of model curve shape parameters (b0, b1, b2, b3, b4 and knot), number of regressors and flexibility selection criterion (RSS, AIC, AICc and BIC) for linear and non-linear models for milk yield in Murciano-Granadina goats; Table S4: Summary of descriptive statistics for milk yield (Kg) in the Murciano-Granadina goat breed; Table S5: Bayesian inference Posterior Mean Distribution for Pearson's correlations between predicted values of curve shape parameters (b0, b1, b2, b3, b4 and knot) across models.

**Author Contributions:** Data curation, F.J.N.G. and J.M.L.J.; Formal analysis, M.G.P.I., F.J.N.G. and V.L.; Funding acquisition, J.V.D.B.; Investigation, M.G.P.I., F.J.N.G., J.M.L.J., J.V.D.B. and M.d.A.M.M.; Methodology, F.J.N.G. and J.V.D.B.; Project administration, J.V.D.B.; Resources, F.J.N.G., J.M.L.J., J.F.A. and M.d.A.M.M.; Software, M.G.P.I., F.J.N.G. and J.M.L.J.; Supervision, F.J.N.G., J.V.D.B. and M.d.A.M.M.; Visualization, V.L.; Writing—original draft, M.G.P.I. and F.J.N.G.; Writing—review & editing, F.J.N.G., V.L., J.M.L.J., J.F.A. and M.d.A.M.M. All authors have read and agreed to the published version of the manuscript.

**Funding:** This research received no external funding.

**Acknowledgments:** The authors want to thank the support and assistance of the National Association of Breeders of Murciano-Granadina Goat Breed, Fuente Vaqueros (Spain) and the PAIDI AGR 218 research group.

**Conflicts of Interest:** The authors declare no conflict of interest.

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
