# Peer review of "Software-Automatized Individual Lactation Model Fitting, Peak and Persistence and Bayesian Criteria Comparison for Milk Yield Genetic Studies in Murciano-Granadina Goats"

_mathematics, doi:10.3390/math8091505_

Round 1

Reviewer 1 Report

Dear authors,

This research is particularly important because it was conducted on goats. Namely, it is known that goat production in the world, and even in many European countries (especially in the Mediterranean) is often performed in (semi) extensive conditions where the possibilities of monitoring and controlling milk production are quite limited. So, the research topic is interesting and actual, the goal and purpose of the research are clearly set, and the results are well explained and compared with the available current literature. So, I reccomend publication.

I am only interested in why on the surveyed farms the milk yield data of goats were collected in so many different ways (AT4, AT4T, AT4M, A6, AT6M, and AT6T)?

Author Response

Dear authors,

This research is particularly important because it was conducted on goats. Namely, it is known that goat production in the world, and even in many European countries (especially in the Mediterranean) is often performed in (semi) extensive conditions where the possibilities of monitoring and controlling milk production are quite limited. So, the research topic is interesting and actual, the goal and purpose of the research are clearly set, and the results are well explained and compared with the available current literature. So, I reccomend publication.

Response: We thank the reviewer for his/her kind comments.

I am only interested in why on the surveyed farms the milk yield data of goats were collected in so many different ways (AT4, AT4T, AT4M, A6, AT6M, and AT6T)?

Response: As it is stated in the manuscript, the milk control routines depended on the habits of the farmers. We sampled animals at 28 farms in the south of Spain, hence, it can easily happen for owners of the facilities to differ in regards their milking routines.

Reviewer 2 Report

Dear Authors,

Please find below some rather global comments.
I hope they are useful.

Introduction

General:

  • Please try to make your line of thoughts clearer. Currently the introduction is not obvious to follow.
  • Simplify strongly your sentences and ideas. Example L42-49: 
    "Contextually, when the objectives of such research .... provided the extrapolation of missing data after the observations of individuals that
    are present." - This sentence is 7 lines long and includes many different ideas. Also, I am not a native speaker, but I find your English is very difficult to follow. 
  • Please be more prudent when arguing. An example L95-97: "... when the specific skill to handle them are not available ...". Mixed models are standard for many quantitative geneticists and most software is not really difficult to use. A good argument would have been that mixed models are by definition linear.... The really interesting aspect of mixed models is that they can combine population and animal specific effects. Also the use of random regression effects protects again overfiting with few data points for a given lactation (e.g.,  https://doi.org/10.3168/jds.S0022-0302(04)73351-2). Anyway the scientific reasoning should here be improved.  
  • Please check your references (and numbers). Example paper [7] is mentioning mixed model, indeed BUT not lactation curves! You should find many other publication that could replace this one.

M&M

  • The range of ages are rather large, more details on their distribution could help.
  • Please be more precise and focused. Example: 2.2. : I do not understand why you explain how total milk yield or normalized yield at 210 DIM (MP210) are computed, but they are not used later.
  • I miss more details on the data you are using, i.e. a figure showing the evolution of the average milk yield. This could also be in Results. Currently but there are only overall results in table S4. Suggestion: mean MY per week in lactation. With 210 days this gives you 30 data points and a way to represent the evolution of MY throughout the lactation. 
  • Please be careful when you compare (2.7) peak and persistency measures. You suggest in L338-341 a method to compute peak and persistency in an harmonized way. Given the paramount importance of definition for these parameters it would use this methods for all models. 
  •  I think that the shape parameters my not be as comparable as you suggest. You should be more prudent (L260-261)

Results

  • More descriptive statistics (see M&M) please.

Discussion

  • I think L509 ff is very relevant. Please strengthen the discussion in this direction. I think you lack also some discussions about the potential differences between goats and cattle. I give an example. If I interpret correctly, some models as MILKBOT had very low % of
    successfully fitted lactation curves. This method behave however very well in cow studies (https://doi.org/10.3168/jds.2011-4905).

Author Response

Reviewer 2

Dear Authors,

Please find below some rather global comments.
I hope they are useful.

Introduction

General:

  • Please try to make your line of thoughts clearer. Currently the introduction is not obvious to follow.

Response: Introduction was rewritten to improve readability and clarify the scope of the paper.

  • Simplify strongly your sentences and ideas. Example L42-49: 
    "Contextually, when the objectives of such research .... provided the extrapolation of missing data after the observations of individuals that
    are present." - This sentence is 7 lines long and includes many different ideas. Also, I am not a native speaker, but I find your English is very difficult to follow. 

Response: The whole paper was checked and revised by a Cambridge ESOL examination instructor to improve readability and prevent the occurrence of grammar mistakes and typos.

  • Please be more prudent when arguing. An example L95-97: "... when the specific skill to handle them are not available ...". Mixed models are standard for many quantitative geneticists and most software is not really difficult to use. A good argument would have been that mixed models are by definition linear.... The really interesting aspect of mixed models is that they can combine population and animal specific effects. Also the use of random regression effects protects again overfiting with few data points for a given lactation (e.g.,  https://doi.org/10.3168/jds.S0022-0302(04)73351-2). Anyway the scientific reasoning should here be improved.  

Response: We added a paragraph to improve scientific reasoning and the citation recommended by the reviewer.

  • Please check your references (and numbers). Example paper [7] is mentioning mixed model, indeed BUT not lactation curves! You should find many other publication that could replace this one.

Response: The aim of this citation was to support the application of SPSS software rather than the application of mixed models. In line with the suggestion by the reviewer, we added the following two citations:

  1. Murphy, M.D.; O’Mahony, M.J.; Shalloo, L.; French, P.; Upton, J. Comparison of modelling techniques for milk-production forecasting. Journal of Dairy Science 2014, 97, 3352-3363, doi:https://doi.org/10.3168/jds.2013-7451.
  2. Mayeres, P.; Stoll, J.; Bormann, J.; Reents, R.; Gengler, N. Prediction of Daily Milk, Fat, and Protein Production by a Random Regression Test-Day Model. Journal of Dairy Science 2004, 87, 1925-1933, doi:10.3168/jds.S0022-0302(04)73351-2.

M&M

  • The range of ages are rather large, more details on their distribution could help.

Response: Information was provided.

  • Please be more precise and focused. Example: 2.2. : I do not understand why you explain how total milk yield or normalized yield at 210 DIM (MP210) are computed, but they are not used later.

Response: After lactation yields were computed, they were standardized/normalized to provide a reasonably equitable comparison of dairy goats with different lactation characteristics as suggested in Norman, et al. [14].

Norman, H.; Cooper, T.; Ross, J., FA. State and national standardized lactation averages by breed for cows calving in 2010; Animal Improvement Programs Laboratory, Agricultural Research Service, USDA: Beltsville, MD, 2010.

  • I miss more details on the data you are using, i.e. a figure showing the evolution of the average milk yield. This could also be in Results. Currently but there are only overall results in table S4. Suggestion: mean MY per week in lactation. With 210 days this gives you 30 data points and a way to represent the evolution of MY throughout the lactation. 

Response: Figure suggested by the reviewer was added.

  • Please be careful when you compare (2.7) peak and persistency measures. You suggest in L338-341 a method to compute peak and persistency in an harmonized way. Given the paramount importance of definition for these parameters it would use this methods for all models. 

Response: There was a misunderstanding which we clarified in text. Peak and persistence were computed following the specific information for each model described in literature, using the same method for all models would be inappropriate. For the functions for which a method to compute such parameters had not been already described we solved them mathematically, that is using the mathematical solution to find local maxima and decreasing stages afterwards.

  •  I think that the shape parameters my not be as comparable as you suggest. You should be more prudent (L260-261)

Response: We agree. The correlations between curve shape parameters (b0, b1, b2, b3 and b4), are presented in Table 3. A moderate evidence for the correlation between b0 and b1 and b3 was suggested by Pearson correlation Bayesian inference analysis (Table 3). However, the values for these correlations were negative and low. Higher positive and negative correlations were anecdotally evidenced between b2 and b4, respectively.

Results

  • More descriptive statistics (see M&M) please.

Response: Figure suggested by the reviewer was added.

Discussion

  • I think L509 ff is very relevant. Please strengthen the discussion in this direction. I think you lack also some discussions about the potential differences between goats and cattle. I give an example. If I interpret correctly, some models as MILKBOT had very low % of successfully fitted lactation curves. This method behave however very well in cow studies (https://doi.org/10.3168/jds.2011-4905).

Response: We added the following section to fulfil the reviewer’s suggestion. “Additionally, our results contrast those found in literature for other species. For instance, the biological idiosyncrasies (time to peak, peak yield, number of peaks across lactation, total yield, value of persistence, among others) of the lactation of the different dairy species may condition the best fitting properties of certain models over the rest. For instance, cattle for which good outputs have been reported when MILKBOT model was fitted [1], sheep and donkeys for which WOOD model has been suggested to fit well [2] and camels [3] for which the best model fit was found for fourth order polynomials. Still, papers usually test the fitness ability of reduced numbers of models that have commonly been used in literature. Hence, no direct comparison can possibly be made, as even if such models perform well, there is a lack of evidence of other potential model reporting better or worse results.”

Reviewer 3 Report

The study of Pizarro Inostroza et al. is very interesting, well designed and written. I only have minor suggestions to the authors.

  1. The introduction is extended, ant he point - purpose of the study is lost. I suggest you reduce the introduction substantially.
  2. The manuscript has no hypothesis. Instead, the list of objectives (Lines 104- 113) are more like materials and methods. For example, the first objective is to provide the syntax for SPSS, but it is indeed included in Materials and methods (Table1). If it was objective of the study, it should have been part of the Results. The actual objective of the study needs to be defined along with the hypothesis.
  3. In Materials and methods there is too many information about statistics used. I suggest reducing the information. The majority of the readers do know how the R2 is calculated or what is the difference between R2 and adj R2. This is not a dissertation but a research paper.
  4. I would suggest to involve the terms of accuracy and precision in your model evaluation. Please, see the paper of Tedeschi (2006; Agricultural Systems 89 (2006) 225–247) about this concept. Moreover, in this paper are defined other statistics, such as CCC that can be used as an indicator of accuracy and precision.
  5. Even though you are working on lactation curves there is not even one figure in your manuscript. I think developing a couple of figures from the best equations it will increase the visibility of the manuscript.

Minor comments:

Line 66: please provide author’s name before [] as in line 174

Author Response

Reviewer 3

Comments and Suggestions for Authors

The study of Pizarro Inostroza et al. is very interesting, well designed and written. I only have minor suggestions to the authors.

  1. The introduction is extended, ant he point - purpose of the study is lost. I suggest you reduce the introduction substantially.

Response: Introduction was reduced and rewritten to improve readability and clarify the scope of the paper as suggested y both reviewers.

  1. The manuscript has no hypothesis. Instead, the list of objectives (Lines 104- 113) are more like materials and methods. For example, the first objective is to provide the syntax for SPSS, but it is indeed included in Materials and methods (Table1). If it was objective of the study, it should have been part of the Results. The actual objective of the study needs to be defined along with the hypothesis.

Response: Objectives of the manuscript were clarified and rewritten.

  1. In Materials and methods there is too many information about statistics used. I suggest reducing the information. The majority of the readers do know how the R2 is calculated or what is the difference between R2 and adj R2. This is not a dissertation but a research paper.

Response: M&M section was reduced by 240 words, following the suggestion made by the reviewer.

  1. I would suggest to involve the terms of accuracy and precision in your model evaluation. Please, see the paper of Tedeschi (2006; Agricultural Systems 89 (2006) 225–247) about this concept. Moreover, in this paper are defined other statistics, such as CCC that can be used as an indicator of accuracy and precision.

Response: The reviewer suggestion was followed. As suggested by Tedeschi [4], the concordance correlation coefficient (CCC), measures if predicted values are precise and accurate. CCC is also known as reproducibility index. According to the same authors [4], CCC is based on the Pearson’s correlation coefficient estimate (r), which measures precision. Bayesian inference Pearson correlation function was used to characterize the posterior distribution of the linear correlation between predicted curve shape parameters (b0, b1, b2, b3 and b4) across models.

  1. Even though you are working on lactation curves there is not even one figure in your manuscript. I think developing a couple of figures from the best equations it will increase the visibility of the manuscript.

Response: We added two Figures following the reviewer suggestion. Graphical representation of the evolution of milk yield (Kg) through lactation from first to ninth lactation in Murciano-Granadina and Graphical representation of the evolution of milk yield (Kg) and Ali and Schaeffer (ALISCH) model fit in Murciano-Granadina, displaying how ALISCH model fitted respecting to days in milk.

Minor comments:

Line 66: please provide author’s name before [] as in line 174

Response: Name was provided.

Round 2

Reviewer 2 Report

Dear authors,

Thank you for your revisions.
Unfortunately they were not highlighted, but I did an electronic comparison of the pdf files.

I have only one major concern:

Could you please change Figure 1 to something more easy to understand. 
It looks as if "average / (day in milk * lactation)" is not very representative and no tendency is visible. 

I try to compare to Figure 2. Here a rather clear tendency is visible. If I understand well the blue line correspond to the overall "average / day in milk".  But this does not looks very plausible given Fig1. Please clarify!

I come back to this suggestion from the first review:

"Compute mean MY per week in lactation. With 210 days this gives you 30 data points and a way to represent the evolution of MY throughout the lactation."

You should also group lactations in Fig1. maybe 1, 2, 3+ or similar.

Still Figure 1 as it is looks strange, compared to Figure 2. I even creates doubts about your data.

Some minor remarks:

L134-135 "... milk controls (AT4, AT4T, AT4M, A6, AT6M, and AT6T) performed at each farm were strictly dependent on farmer’s habits"  sounds really strange. I suggest this wording: "... milk performance recordings were performed at each farm according to the ICAR protocol (AT4, AT4T, AT4M, A6, AT6M, or AT6T) chosen by the farmer."

L139-147: I still have problems to understand why this whole part (in fact whole 2.2). Why you compute RP, NP or MP210? You never use it later (or do I miss something?). I would avoid defining things that will be never used. I think you simply use observed milk yields in your models. Correct? If you insist please keep it, but I see no benefit in doing so.

If I keep an animal science point of view I do not agree with this reply:
"Response: There was a misunderstanding which we clarified in text. Peak and persistence were computed following the specific information for each model described in literature, using the same method for all models would be inappropriate. For the functions for which a method to compute such parameters had not been already described we solved them mathematically, that is using the mathematical solution to find local maxima and decreasing stages afterwards."

My point is that many animal science papers have shown that even if it is always called "persistency" it is not the same trait. If you add this disclaimer somewhere in 2.7, As in line 338: "[28] . It has to be noted that theoretical definitions of persistency can be very different, but we decided to stay as close as possible to the definitions associated to the tested functions. Therefore, as Table S2 suggests, persistency could be computed differently... " I agree that you keep your way of reasoning. A quick internet search should allow you to find appropriate references.

Author Response

Comments and Suggestions for Authors

Dear authors,

Thank you for your revisions.

Unfortunately they were not highlighted, but I did an electronic comparison of the pdf files.

Response: Sorry, we enclosed the file with tracking changes in the zip file, but there may have been a problem while uploading.

I have only one major concern:

Could you please change Figure 1 to something more easy to understand. 
It looks as if "average / (day in milk * lactation)" is not very representative and no tendency is visible. 

I try to compare to Figure 2. Here a rather clear tendency is visible. If I understand well the blue line correspond to the overall "average / day in milk".  But this does not looks very plausible given Fig1. Please clarify!

I come back to this suggestion from the first review:

"Compute mean MY per week in lactation. With 210 days this gives you 30 data points and a way to represent the evolution of MY throughout the lactation."

You should also group lactations in Fig1. maybe 1, 2, 3+ or similar.

Still Figure 1 as it is looks strange, compared to Figure 2. I even creates doubts about your data.

Response: The figure suggested by reviewer has been created. We have decided to group from 4th lactation on rather than 1, 2 3 or more to balance the figure.

Some minor remarks:

L134-135 "... milk controls (AT4, AT4T, AT4M, A6, AT6M, and AT6T) performed at each farm were strictly dependent on farmer’s habits"  sounds really strange. I suggest this wording: "... milk performance recordings were performed at each farm according to the ICAR protocol (AT4, AT4T, AT4M, A6, AT6M, or AT6T) chosen by the farmer."

Response: We followed reviewer’s suggestion.

L139-147: I still have problems to understand why this whole part (in fact whole 2.2). Why you compute RP, NP or MP210? You never use it later (or do I miss something?). I would avoid defining things that will be never used. I think you simply use observed milk yields in your models. Correct? If you insist please keep it, but I see no benefit in doing so.

Response: As it is said in text lactation yields were normalized to 210 days, that is why we prefer to keep it.

If I keep an animal science point of view I do not agree with this reply:
"Response: There was a misunderstanding which we clarified in text. Peak and persistence were computed following the specific information for each model described in literature, using the same method for all models would be inappropriate. For the functions for which a method to compute such parameters had not been already described we solved them mathematically, that is using the mathematical solution to find local maxima and decreasing stages afterwards."

My point is that many animal science papers have shown that even if it is always called "persistency" it is not the same trait. If you add this disclaimer somewhere in 2.7, As in line 338: "[28]. It has to be noted that theoretical definitions of persistency can be very different, but we decided to stay as close as possible to the definitions associated to the tested functions. Therefore, as Table S2 suggests, persistency could be computed differently... " I agree that you keep your way of reasoning. A quick internet search should allow you to find appropriate references.

Response: We agree with the point made by the reviewer and think it is important to add the section proposed.